# Instruction-tuning Aligns LLMs to the Human Brain

**Khai Loong Aw**[*], **Syrielle Montariol**[†], **Badr AlKhamissi**[†]
**Martin Schrimpf**[‡], **Antoine Bosselut**[‡]
EPFL
awkhailoong@gmail.com

## Abstract

Instruction-tuning is a widely adopted finetuning method that enables large language models (LLMs) to generate output that more closely resembles human responses. However, no studies have shown that instruction-tuning actually teaches LLMs to process language in a similar manner as humans. We investigate the effect of instruction-tuning on aligning LLM and human language processing mechanisms in two ways: (1) *brain alignment*, the similarity of LLM internal representations to neural activity in the human language system, and (2) *behavioral alignment*, the similarity of LLM and human behavior on a reading task. We assess 25 vanilla and instruction-tuned LLMs on three datasets involving humans reading naturalistic stories and sentences, and find that instruction-tuning generally enhances brain alignment (∼6%), but has no similar effect on behavioral alignment. To identify factors underlying this improvement in brain alignment, we compute correlations between brain alignment and various LLM properties, such as model size, problem-solving, and world knowledge understanding. Notably, we find a strong positive correlation between brain alignment and model size (r = 0.95), as well as performance on tasks requiring world knowledge (r = 0.81). Our results demonstrate that instruction-tuning LLMs improves both world knowledge representations and brain alignment, suggesting that the mechanisms that encode world knowledge in LLMs also improve representational alignment to the human brain.

## 1   Introduction

Instruction-tuning is a widely adopted method that finetunes large language models (LLMs) on datasets containing many examples of different tasks and their descriptive instructions, enhancing their ability to generalize to previously unseen tasks by learning to follow provided instructions (Wang et al., 2022c). Despite costing only a small fraction of compute relative to pretraining (Chung et al., 2022), instruction-tuning yields large performance improvements on reasoning and problem-solving benchmarks. This generalization has allowed LLMs to tackle open-world reasoning tasks previously achievable only by humans (Zhang et al., 2023) while only using a few (or zero) task-specific training examples.

In addition to teaching LLMs to understand and follow human instructions, instruction-tuning also improves the ability of LLMs to mimic human-written ground-truth outputs. This fluency allows them to produce more controllable and predictable output that is deemed (1) more desirable to human evaluators (Zhang et al., 2023; Chung et al., 2022; Wang et al., 2022b), (2) more aligned to human values (Chia et al., 2023), and (3) more stylistically similar to human outputs (Dasgupta et al., 2022; Safdari et al., 2023).

Consequently, instruction-tuning yields LLMs more similar to humans in both capability and output resemblance. From a neuroscience perspective, this begs the question: **Does instruction-tuning make LLMs more functionally similar to the human language system?**

---

[*]Work was done during internship at EPFL
[†]Equal contribution
[‡]Equal supervision / senior authors

Previous work has shown that models with higher task performance are more aligned to the human language system (Schrimpf et al., 2021; Goldstein et al., 2022; Caucheteux & King, 2022), hitting the estimated noise ceiling[1] on some datasets. However, there has been no similar study on how instruction-tuning affects alignment with the human language system.

In this work, we measure the effect of instruction-tuning on the alignment between language mechanisms in LLMs and the human brain in two ways: (1) *brain alignment*, an "internal" metric that assesses how well internal feature representations of LLMs match neural representations in the human language system, and (2) *behavioral alignment*, an "external" metric which evaluates the similarity between LLM and human behavioral measurements. In our study, both LLMs and humans are presented with the same language stimuli comprised of naturalistic stories and sentences. For LLMs, we record their internal representations and per-word perplexity. For humans, we use previously recorded brain activity data from functional magnetic resonance imaging (fMRI) experiments and per-word reading times.

To measure brain alignment, we use the Brain-Score (Schrimpf et al., 2018) linear predictivity metric, assessing how well LLM representations predict human brain activity in response to the same language stimuli (Jain & Huth, 2018; Toneva & Wehbe, 2019; Schrimpf et al., 2021; Oota et al., 2023), using data from three neural datasets: Pereira et al. (2018), Blank et al. (2014), and Wehbe et al. (2014). To evaluate behavioral alignment, we use a benchmark in Brain-Score which calculates the Pearson correlation between LLM per-word perplexity and human per-word reading times from the Futrell et al. (2018) dataset. Perplexity for LLMs and reading times for humans offer insights into comprehension difficulty (Ehrlich & Rayner, 1981; Hale, 2001; Smith & Levy, 2013), allowing us to examine whether LLMs match humans in their patterns of words and sentences they find challenging or surprising. Because models vary in their brain and behavioral alignment across different architectures and training objectives (Schrimpf et al., 2021), we estimate the effect of instruction-tuning by evaluating 8 vanilla LLMs and 17 LLMs that were further instruction-tuned, and report a significant increase in brain alignment due to instruction-tuning.

To investigate *why* instruction-tuning increases alignment to human brain activity, we study the relationships between brain alignment and various LLM properties. Specifically, we compute Pearson correlations between an LLM's brain alignment and its properties, including next-word prediction (NWP) ability, model size, a range of problem-solving abilities, and world knowledge spanning different domains. We evaluated the latter two properties using the Big-Bench Hard benchmark (BBH; Suzgun et al., 2022) and the Massive Multi-task Language Understanding benchmark (MMLU; Hendrycks et al., 2021), respectively.

We report three major findings:

1. Instruction-tuning generally aligns LLM representations to human brain activity, increasing brain alignment by 6.2% on average (Figure 1).

2. Investigating the factors underlying LLM-brain alignment, we find that brain alignment is strongly correlated with world knowledge (r = 0.81) and model size (r = 0.95) (Figure 2).

3. Surprisingly, our results indicate that instruction-tuning LLMs generally does not enhance behavioral alignment with human reading times. Furthermore, behavioral alignment on this dataset is poorly correlated with all other measures we investigate, including task performance and model size (Figure 3).

## 2 Background & Related Work

**Effect of Instruction-tuning on LLMs.** Instruction-tuning is an effective method for enhancing LLM capability and controllability. It trains LLMs using pairs of human instructions and desired outputs. The benefits of instruction-tuning can be categorized into three key aspects (Zhang et al., 2023): (1) it bridges the disparity between the pretraining objective

---

[1]In fMRI recordings, a noise ceiling for representational similarity can be computed by sampling from the same participant twice, obtaining an upper limit for how well an ideal model could perform, defined by the internal consistency of neural responses and noise level of the data gathering process.

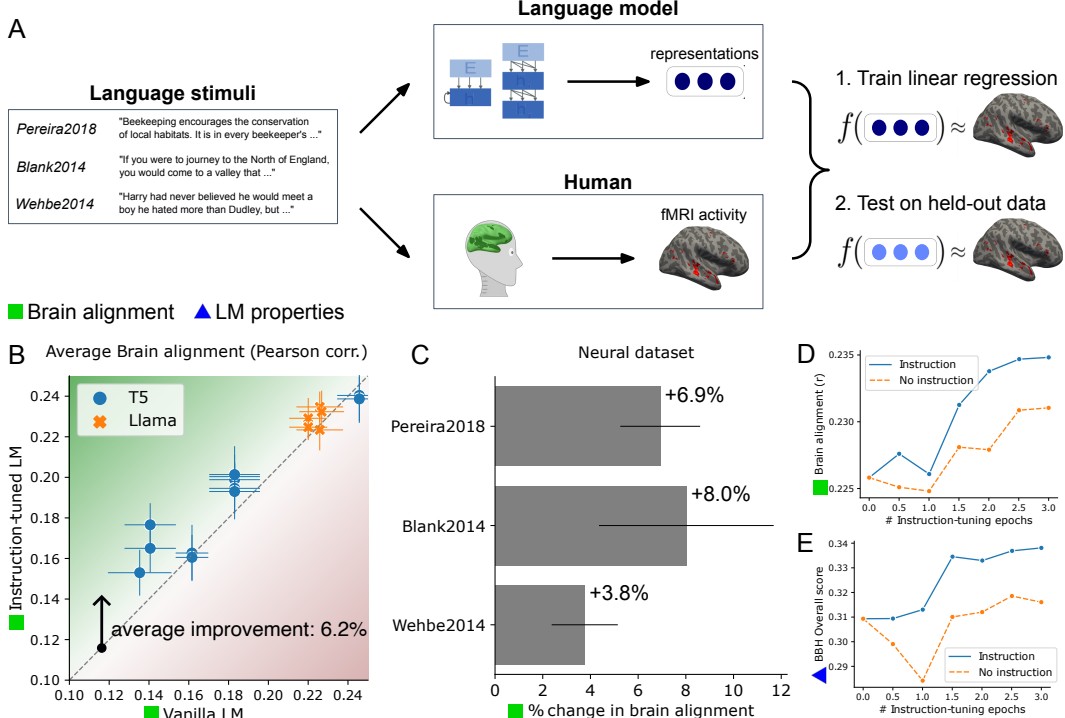

Figure 1: **Instruction-tuning aligns LLM representations to human brain activity. (A)** The same language stimuli are presented to LLMs and human participants. Next, we fit a linear regression model from LLM layer activations to fMRI responses in the human language system. We apply this linear model to predict held-out fMRI responses from the original corpus of recordings, and compute brain alignment as the Pearson correlation between the predicted and actual fMRI responses. We evaluate 25 vanilla and instruction-tuned LLMs with sizes between 77M and 33B parameters. We compute the average across 3 neural datasets of humans reading naturalistic stories and sentences: PEREIRA2018, BLANK2014, and WEHBE2014. **(B)** Instruction-tuning improves average brain alignment by 6.2% on average. Each point above the identity line is an instruction-tuned LLM with greater brain alignment than its vanilla version. Error bars, here and elsewhere, represent median absolute deviation over human participants. **(C)** Instruction-tuning improves average brain alignment on all three datasets. **(D)** We instruction-tune LLaMA-7B on the Alpaca dataset ("Instruction" model) and train an ablation model with the same data, but without the instruction in each training sample ("No Instruction" model). Our results show that brain alignment improvements are due to both (1) training data (present in both models) and (2) training LLMs to understand and follow instructions (present only in the first model).

of LLMs (next-word prediction) and the goal of accurately following human instructions, (2) it achieves greater control and predictability of model behavior compared to standard LLMs, allowing researchers to make them more similar to humans in both capability and output similarity (Chia et al., 2023; Dasgupta et al., 2022; Safdari et al., 2023), and (3) it often costs only a small fraction of compute relative to pretraining, enabling LLMs to swiftly adapt to target domains (Chung et al., 2022). We contribute to this research area from a neuroscience perspective, by studying whether instruction-tuning makes LLMs more aligned to the human language system in terms of brain and behavioral alignment.

**Effect of Finetuning on Brain alignment.** Prior works have studied how finetuning affects LMs' alignment to human brain activity. These include finetuning on a wide range of downstream NLP tasks (Oota et al., 2022), finetuning to summarize narratives (Aw & Toneva, 2023), and finetuning to directly predict brain activity recordings (Schwartz et al., 2019). These studies aim to use brain alignment to study how finetuning affects LMs and their representations. Our work builds on this line of research by demonstrating that

| Instruction | Input | Output |
|---|---|---|
| "Write a short paragraph about the given topic." | "The importance of using renewable energy." | "The use of renewable energy is growing..." (paragraph) |

Table 1: **Example of Instruction-tuning training data format: (Instruction, Input, Output)** from the Alpaca dataset (Taori et al., 2023). The input field is optional for certain instructions.

instruction-tuning aligns LLM representations to human brain activity. We also investigate why instruction-tuned LLMs align to brain activity by testing the correlation of brain alignment with various world knowledge domains and problem-solving abilities.

**LM properties linked to Brain alignment.** One exciting research area focuses on disentangling the contribution of various LM properties towards brain alignment. These include studying how brain alignment is driven by next-word prediction ability (Schrimpf et al., 2021; Caucheteux & King, 2022), multi-word semantics (Merlin & Toneva, 2022), performance on various NLP tasks (Oota et al., 2022), and model size (Antonello et al., 2023). To contribute to this growing body of work, we use instruction-tuned LLMs. They are especially useful as they have been trained to respond to a standard question-answer format, allowing us to evaluate LLMs on a wide array of tasks and in a more fine-grained manner. We expand this area of research by demonstrating that world knowledge is a key property underlying LLM-brain alignment.

## 3 Language Models

We evaluate the brain alignment of 25 large language models (LLMs) from two model families: T5 (Raffel et al., 2020) and LLaMa (Touvron et al., 2023a). T5 models are encoder-decoder LLMs pre-trained using a masked infilling objective on the Colossal Common Crawl Corpus (C4), a corpus of 356 billion tokens, and then further finetuned on a multi-task mixture of unsupervised and supervised tasks converted into a text-to-text format. We use all five T5 models, with sizes between 77M to 11B. LLaMA models (Touvron et al., 2023a) are decoder-only LLMs trained on 1.6 trillion tokens from a mixture of corpora including C4, English CommonCrawl, Wikipedia, GitHub. We use the 7B, 13B, and 33B versions.

For the instruction-tuned variants of T5 models, we use a variety of models finetuned on FLAN (15M examples for 1,836 different tasks accompanied by instructions, Chung et al., 2022), Alpaca (52K instruction-following examples generated with methods inspired by Self-Instruct, Wang et al. (2022a), Taori et al., 2023), and GPT4ALL (437K instruction-following examples generated with GPT-3.5-turbo, Anand et al., 2023) datasets. For the instruction-tuned variants of LLaMa, we use Vicuna's 7B, 13B, and 33B models (Chiang et al., 2023), which are finetuned on user-shared conversations. We also use StableVicuna-13B, which further refines Vicuna-13B using reinforcement learning from human feedback (RLHF) (Ouyang et al., 2022) on a range of conversational and instructional datasets. We also use the 7B version of Alpaca (Taori et al., 2023). We provide more details in Appendix A.

We also tested 8 models from the GPT2 family; four vanilla models (Small, Medium, Large, XL) and their corresponding versions further instruction-tuned on Alpaca. However, the models performed close to random on reasoning and world knowledge benchmarks (BBH, MMLU). We hypothesize this is due to their size (all < 1.5B parameters), making fine-tuning on a relatively small instruction-following dataset insufficient. Moreover, GPT2 was not originally pretrained to perform instruction-following tasks (unlike T5 models). We include their MMLU, BBH, brain and behavioral alignment results in Appendix G, H, and L.

## 4 Brain Alignment

Brain alignment refers to the method of evaluating the similarity between LLM representations and human brain activity (Figure 1A and Appendix C). This relies on fMRI recordings of human subjects while they read language stimuli on potentially any topic (here: Pereira

et al., 2018; Blank et al., 2014; Wehbe et al., 2014). The same language stimuli from prior brain recordings are provided as input to LLMs, whose intermediate layer activations are extracted as their representations of the language stimuli. We follow a general approach previously used in several works (Schrimpf et al., 2018; Jain & Huth, 2018; Toneva & Wehbe, 2019; Schrimpf et al., 2021; Oota et al., 2023; Aw & Toneva, 2023). Specifically, we use the linear predictivity metric implemented in Brain-Score (Schrimpf et al., 2018), which fits a linear regression model from LLM layer activations to fMRI responses in the human language system. We then apply this linear model to predict held-out fMRI responses from the original corpus of recordings, and compute brain alignment as the Pearson correlation between the predicted and actual fMRI responses. For each LLM, we evaluate its brain alignment for every layer (e.g., LLaMA-7B has 32 layers), and use the highest value as the LLM's brain alignment value, following Schrimpf et al. (2018).

**Datasets** We use three fMRI datasets to measure the brain alignment of LLMs. Each dataset involves a different set of human participants and a different set of language stimuli.

PEREIRA2018 (experiments 2 and 3 from Pereira et al., 2018): In experiment 2, 9 participants read 384 sentences taken from 96 text passages. In experiment 3, 6 participants read 243 sentences from 72 text passages. Each sentence was displayed for 4 seconds on a screen.

BLANK2014 (Blank et al., 2014): The data consists of fMRI recordings of 5 human participants listening to 8 naturalistic stories from the Natural Stories Corpus (Futrell et al., 2018).

WEHBE2014 (Wehbe et al., 2014): The data includes fMRI recordings of 8 human participants reading chapter 9 of the book *Harry Potter and the Sorcerer's Stone* (Rowling et al., 1998). Participants read the chapter at a fixed interval of one word every 0.5 seconds.

### 4.1 Instruction-tuning aligns LLM representations to human brain activity

First, we study the effect of instruction-tuning on LLM brain alignment. We compute each LLM's average brain alignment as the mean of its brain alignment on the 3 neural datasets. We find that instruction-tuning improves brain alignment by an average of 6.2% across tested LLMs (Figure 1B). This holds across all three neural datasets, with improvements of +6.9% on PEREIRA2018, +8.0% improvement on BLANK2014, and +3.8% on WEHBE2014 (Figure 1C). Moreover, a smaller instruction-tuned model can attain higher brain alignment than a larger vanilla model from the same family (e.g., Alpaca-7B v.s. LLaMa-13B, see Appendix G). However, some of the larger models (T5-XL, LLaMA) have minimal or no increases, possibly because the vanilla models are already close to the noise ceiling (Appendix G).

Next, to longitudinally study how instruction-tuning aligns LLM representations to brain activity, we separately instruction-tune a LLaMA-7B model on the Stanford Alpaca instruction dataset (Taori et al., 2023) for 3 epochs. By evaluating checkpoints regularly during training, we find that instruction-tuning progressively improves brain alignment (Figure 1D, Appendix K). We also disambiguate the effect on brain alignment of (1) the instruction-following ability gained from instruction-tuning and (2) added training data. We fine-tune LLaMA-7B with the same process and data, but remove the instruction from each training sample. Brain alignment of this ablated model increases during fine-tuning but stays lower than its instruction-following counterpart (Figure 1D), indicating that brain alignment improvements are due to both factors. In Appendix J, we also report preliminary results using recent models from the Gemma (Team et al., 2024) and LLaMA-2 (Touvron et al., 2023b) families and observe similar trends. Further, these trends hold across different similarity metrics besides Linear predictivity, e.g., RSA and CKA (Appendix J).

### 4.2 Factors underlying LLM-brain alignment

To study why LLMs and human brains align in their representations, we compute the Pearson correlation between LLM brain alignment and various properties of LLMs: performance on a benchmark testing various reasoning abilities (BBH; Suzgun et al., 2022), performance on a benchmark testing world knowledge in various domains (MMLU; Hendrycks et al., 2021), language modeling ability, and model size.

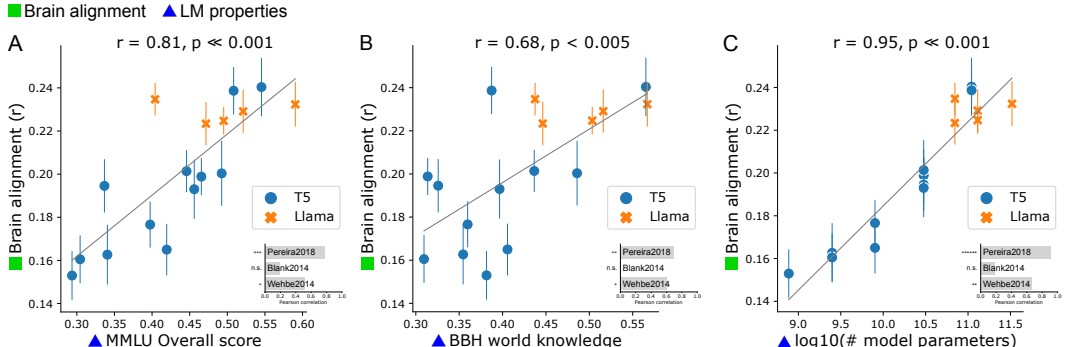

Figure 2: **World knowledge and model size are important factors underlying LLM-brain alignment.** To identify factors underlying brain alignment, we test Pearson correlations between brain alignment and various LLM properties, such as model size, world knowledge in various domains (MMLU benchmark), and various types of problem-solving abilities (BBH benchmark). In the figure, insets display results on individual datasets, with stars reflecting statistical significance (n.s. = p > 0.05, * = p < 0.05, ** = p < 0.005, etc.) **(A)** Brain alignment is significantly and strongly correlated with world knowledge as evaluated by the MMLU Overall score (r = 0.81), which reports the mean performance across all MMLU subjects. **(B)** Brain alignment is significantly and strongly correlated with performance on the world knowledge task category in BBH (r = 0.68). **(C)** Brain alignment is significantly and strongly correlated with model size (logarithm of number of parameters) (r = 0.95). In Appendix I, we provide a larger version of this figure with labels for each data point.

| Benchmark, Task category | Correlation (*r*) to Brain Alignment | Adjusted *p*-value | No. tasks | Avg. Model Accuracy |
|---|---|---|---|---|
| MMLU, Overall Score | **0.809** | **< 0.0005** | 57 | 0.36 |
| MMLU, STEM | **0.792** | **< 0.0005** | 18 | 0.28 |
| MMLU, Humanities | **0.791** | **< 0.0005** | 13 | 0.34 |
| MMLU, Social Sciences | **0.807** | **< 0.0005** | 12 | 0.41 |
| MMLU, Others | **0.809** | **< 0.0005** | 14 | 0.40 |
| BBH, Overall score | 0.384 | 0.18 | 23 | 0.28 |
| BBH, Algorithmic reasoning | 0.194 | 0.56 | 8 | 0.22 |
| BBH, Language understanding | 0.163 | 0.59 | 3 | 0.43 |
| BBH, World knowledge | **0.679** | **< 0.005** | 5 | 0.36 |
| BBH, Multilingual reasoning | -0.035 | 0.90 | 1 | 0.19 |
| BBH, Others | 0.478 | 0.08 | 6 | 0.27 |

Table 2: **Brain alignment strongly correlates with world knowledge in all subject domains in MMLU and the world knowledge category in BBH.** Brain alignment is not significantly correlated with other problem-solving abilities in BBH (e.g., algorithmic or multilingual reasoning). We obtain p-values after performing false discovery rate (FDR) correction.

**MMLU and BBH.** MMLU is designed to measure the world knowledge of LLMs across many subject domains. It contains 57 tasks, categorized into four world knowledge subject domains: STEM, Humanities, Social Sciences, and Others (a broad category ranging from finance to marketing to professional medicine). BBH contains 23 tasks, grouped into four types of problem-solving: Algorithmic and Multi-Step Arithmetic Reasoning; Natural Language Understanding; Use of World Knowledge; and Multilingual Knowledge and Reasoning. For both benchmarks, we follow the category classifications from the original papers. We perform the evaluations using the `instruct-eval` repository[2] with default settings (3-shots for BBH, 5-shots for MMLU) and preset prompts. We measure the Pearson correlation (and its p-value) between LLM-brain alignment and performance in each category of MMLU and BBH. We obtain p-values after false discovery rate (FDR) correction.

---

[2]https://github.com/declare-lab/instruct-eval

**World Knowledge.** We find that brain alignment is significantly and strongly correlated with world knowledge. Brain alignment is highly correlated with the MMLU Overall score (r = 0.81, p < 0.001, Figure 2A), which reports the mean performance across all world knowledge subject domains on MMLU. Similarly, brain alignment is also strongly correlated with performance on tasks in the world knowledge category of BBH (r = 0.68, p < 0.005; Figure 2B). Notably, brain alignment is poorly correlated with all other dimensions of BBH (see Table 2), though this could also be due to limitations of the tested models, as indicated by their low raw performance scores on some tasks. Overall, our results provide a strong signal that more accessible representations of world knowledge are a key factor in aligning LLM representations to human brain activity.

**Language Modeling Ability.** Prior works have shown correlations between brain alignment and next-word prediction (NWP) ability (Caucheteux & King, 2022; Schrimpf et al., 2021). We find similar results for correlation between brain alignment and NWP loss (r = -0.54, p < 0.05, Appendix I). Interestingly, the strength of the correlation is weaker than that between brain alignment and world knowledge performance (r = 0.81). This suggests that world knowledge understanding is a better predictor of brain alignment than NWP ability.

**Model Size.** Finally, we find that brain alignment is significantly and strongly correlated with model size (r = 0.95, p < 0.001, Figure 2C), as measured by the logarithm of the number of model parameters. Schrimpf et al. (2021) observe such a pattern for language models, and we find the pattern holds for instruction-tuned models, and models trained at a larger scale than their study (7B+ parameters). However, model size alone does not determine brain alignment. Our results show that smaller instruction-tuned LLMs can have greater brain alignment than larger vanilla models. For example, LLaMA-13B obtains brain alignment of 0.220, Vicuna-13B obtains 0.229, LLaMA-33B obtains 0.227, and Vicuna-33B obtains 0.232. Hence, Vicuna-13B has greater brain alignment than LLaMA-33B, due to instruction-tuning, despite being less than 40% its size. We observe a similar trend in another four models: T5-base, Flan-T5-base, T5-large, Flan-T5-large. Also, prior works showed that large random models achieve poor brain alignment (Schrimpf et al., 2021). These results demonstrate there are LLM properties aside from model size that contribute significantly to brain alignment.

**Neural datasets.** Brain alignment is strongly correlated with world knowledge and model size on PEREIRA2018 and WEHBE2014, but less strongly correlated on BLANK2014, possibly because BLANK2014 has fewer participants (N = 5) leading to greater noise in the results.

## 5 Behavioral Alignment

In the previous section, we show that instruction-tuning aligns the internal representations of LLMs to human *brain activity* (Section 4.1). In this section, we explore whether instruction-tuning also aligns LLM behavior to human *behavior*.

Following the approach of Schrimpf et al. (2021) implemented in the Brain-Score package (Schrimpf et al., 2020), we measure behavioral alignment by evaluating the similarity between LLM per-word perplexity and human per-word reading times given the same language stimuli (Appendix D). We use the self-paced reading times dataset from Futrell et al. (2018), consisting of the reading times of 179 human participants recorded while they were visually presented with 10 naturalistic stories. We provide language stimuli from this data as input to LLMs and evaluate behavioral alignment by computing the Pearson correlation between per-word LLM perplexity and per-word human reading times.

One potential concern in the behavioral alignment methodology is that human reading time is influenced by word length; longer words may take more time to read, even if they are not more challenging or surprising. Following Schrimpf et al. (2021), we address this by computing the surprisal of a multi-token word as the sum of the surprisals of its individual tokens. Importantly, we use consistent tokenizers when comparing vanilla LLMs against their instruction-tuned versions, thereby controlling for the effect of word length when evaluating the impact of instruction-tuning on behavioral alignment.

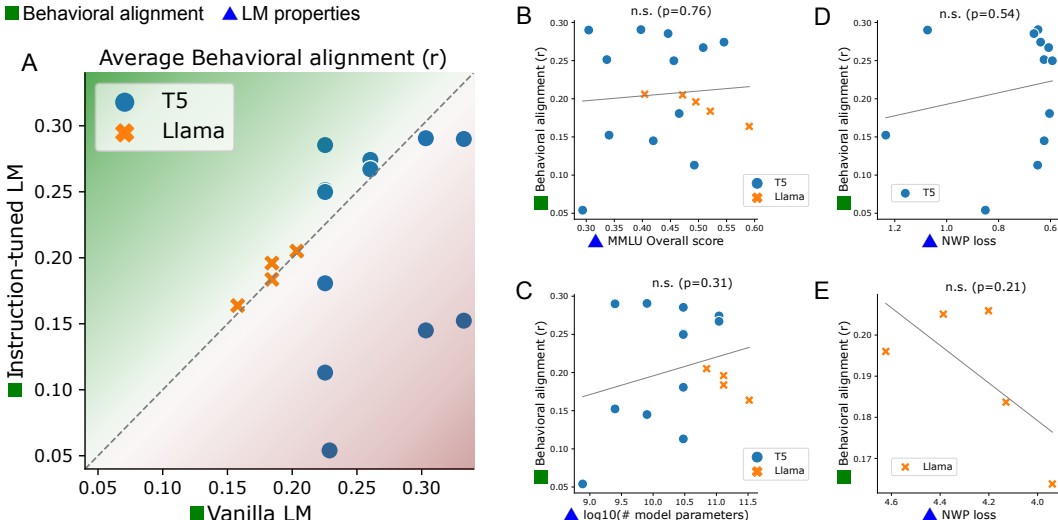

Figure 3: **Instruction-tuning LLMs generally does not improve behavioral alignment to human reading times. Furthermore, behavioral alignment correlates poorly with all other tested measures:** world knowledge, model size, and next-word prediction (NWP) ability. To compute behavioral alignment, we use the FUTRELL2018 benchmark in Brain-Score. The same language stimuli (naturalistic stories) are presented to LLMs and human participants. We then compute the Pearson correlation between per-word LLM perplexity and human reading times as the behavioral alignment. **(A)** Instruction-tuning does not generally improve behavioral alignment. Furthermore, behavioral alignment is poorly and not significantly correlated with all other measures: **(B)** world knowledge (p = 0.76), **(C)** model size (p = 0.31), **(D)** NWP loss for T5 models (p = 0.54), and **(E)** NWP loss for LLaMA models (p = 0.21). In Appendix M, we provide a larger, labeled version of this figure.

## 5.1 Instruction-tuning generally does not improve behavioral alignment

Using the same models as in Section 4, we compare the behavioral alignment of each instruction-tuned LLM against its vanilla counterpart. Our results indicate instruction-tuning generally does not improve behavioral alignment to human reading times (Figure 3A). For half of the LLMs tested, it results in no change or reduced behavioral alignment. Our results align with recent studies (Gao et al., 2023; Steuer et al., 2023) that show improving LLM performance does not necessarily improve alignment with human reading behavior.

## 5.2 Factors underlying behavioral alignment

Next, we test the correlation between LLM behavioral alignment and model size, next-word prediction ability, various reasoning abilities (measured by BBH), and world knowledge across various domains (measured by MMLU). Contrary to our findings on the brain alignment correlations with model size and world knowledge (Section 4.2), we find that LLM behavioral alignment is not correlated with these factors: world knowledge (p = 0.76, Figure 3B), model size (p = 0.31, Figure 3C), next-word prediction loss for T5 models (p = 0.54, Figure 3D), and next-word prediction loss for LLaMA models (p = 0.21, Figure 3E).

## 6 Discussion

### 6.1 Implications for NLP: Building LLMs

**Using brain alignment for LLM interpretability.** Prior works have used human brain activity to interpret models (Dong & Toneva, 2023) and build more performant models (Dapello et al., 2020; Safarani et al., 2021; Dapello et al., 2022). Instruction-tuning has emerged as a breakthrough technique to improve LLM abilities and quality of outputs, and allows LLMs to adapt to new tasks with minimal task-specific training. However,

how instruction-tuning alters LLM internal representations to achieve these improvements remains an open question. Brain activity provides a neuroscientific angle to investigate this question. Our results show that instruction-tuning improves both performance on world knowledge benchmarks and human brain alignment, suggesting that the mechanisms that encode world knowledge in LLMs also align LLM representations to the human brain.

### 6.2 Implications for Neuroscience: Studying LLM-Human Alignment

**Instruction-tuned LLMs are useful for studying LLM properties underlying brain and behavioral alignment.** To identify why LLM and human brains share representational similarities, prior work has mostly focused on high-level properties such as model size (Antonello et al., 2023), as well as external behaviors such as predicting missing words (Schrimpf et al., 2021; Caucheteux & King, 2022). However, a key to understanding these similarities is to identify internal properties of LLMs that underlie brain alignment, including the amount of knowledge LLMs encode, e.g., factual (AlKhamissi et al., 2022) and commonsense (Bosselut et al., 2019; Sap et al., 2020). Our work is the first to show that we can harness instruction-tuned LLMs for this purpose. Because they have been trained to respond to a general instruction format, we can evaluate LLMs on diverse tasks in a fine-grained manner, allowing the study of LLM properties both internal (e.g., knowledge) and external (e.g., behavior), and how they correlate with brain and behavioral alignment.

**World knowledge shapes language comprehension and brain activity.** Our results show that world knowledge is a key factor in LLM brain alignment. LLMs demonstrating greater world knowledge across all tested subject domains have representations more similar to the human brain. This suggests that world knowledge influences human brain activity and shapes the language comprehension systems in our brain.

### 6.3 Limitations and Future Work

**Examining more dimensions of behavior.** Our work and many prior works compare LLM and human next-word surprisal on reading tasks (Wilcox et al., 2020; Schrimpf et al., 2021; Eghbal A. Hosseini et al., 2023), which evaluates only a single behavioral dimension for LLMs (per-word perplexity) and humans (reading times). For the models we tested, behavioral alignment is not significantly correlated with model size, world knowledge, or next-word prediction ability. While next-word prediction performance correlates with alignment to human reading times across many LMs (Schrimpf et al., 2021), this trend does not hold up in recent transformer models (Oh & Schuler, 2023), having a surprising negative correlation with parameter count (Oh et al., 2022). In the future, we hope to evaluate LLMs on a wider range of behavioral dimensions (e.g., van Duijn et al., 2023; Koo et al., 2023; Kauf et al., 2024) to more holistically evaluate LLM-human behavioral alignment.

**Brain alignment datasets with humans performing diverse tasks.** We study brain alignment using neural datasets of humans reading naturalistic stories and sentences in English. It would be interesting to study brain alignment to human participants attempting the BBH and MMLU benchmarks, but this data unfortunately does not exist. This may explain why brain alignment is not significantly correlated with many categories of problem-solving on BBH, e.g., algorithmic reasoning. In the future, we hope to study brain alignment with human participants performing more diverse sets of tasks, e.g., reading computer program code (Ivanova et al., 2020). This can identify more factors underlying LLM-brain alignment, and provide insights into how brain activity and the human language system may be shaped by various forms of problem-solving. Furthermore, some of the larger models exceed the noise ceiling estimates in our neural datasets (Appendix G), highlighting the need for more neural datasets and better ways of computing noise ceiling estimates.

## 7 Conclusion

We investigate whether instruction-tuning improves the alignment of LLMs to the human language system. We evaluate 25 LLMs with parameter sizes ranging from 77 million to 33 billion, across three neural datasets of humans reading naturalistic stories and sentences. We find that instruction-tuning generally improves the alignment of LLM representations

to brain activity. Exploring the factors underlying LLM-brain alignment, we discover that world knowledge and model size are key determinants of brain alignment. This suggests that world knowledge helps shape representations in the human language system, and highlights the importance of integrating world knowledge in developing future LLMs.

## 8 Reproducibility Statement

We provide our full results in Appendices G, H, and L. We consider this important as our experiments may be computationally expensive to replicate. We investigated 25 LLMs, with the largest models having 33B parameters, on many datasets: brain alignment (PEREIRA2018, BLANK2014, WEHBE2014), behavioral alignment (FUTRELL2018), next-word prediction (WikiText-2) and other evaluations (BBH, MMLU).

All models and code repositories used in our study are open-source and their corresponding links are provided in Appendix B and F for full transparency and reproducibility. To calculate brain and behavioral alignment, we used the Brain-Score repository (www.github.com/brain-score/language), a publicly accessible resource for conducting these assessments. We encourage researchers interested in replicating our findings to refer to the provided links and consult the Brain-Score repository for further details on datasets and the evaluation process. To evaluate LLMs on BBH and MMLU, we utilize the widely-used instruct-eval repository (https://github.com/declare-lab/instruct-eval) with default settings.

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

## A  Language Models: Parameter count and Number of Layers

| Model | Parameter Count | Number of Layers |
|---|---|---|
| t5-small | 77 M | 16 |
| flan-t5-small | 77 M | 16 |
| t5-base | 250 M | 24 |
| flan-t5-base | 250 M | 24 |
| flan-alpaca-base | 250 M | 24 |
| t5-large | 800 M | 48 |
| flan-t5-large | 800 M | 48 |
| flan-alpaca-large | 800 M | 48 |
| t5-xl | 3 B | 48 |
| flan-t5-xl | 3 B | 48 |
| flan-alpaca-xl | 3 B | 48 |
| flan-gpt4all-xl | 3 B | 48 |
| flan-sharegpt-xl | 3 B | 48 |
| flan-alpaca-gpt4-xl | 3 B | 48 |
| t5-xxl | 11 B | 48 |
| flan-t5-xxl | 11 B | 48 |
| flan-alpaca-xxl | 11 B | 48 |
| llama-7b | 7 B | 32 |
| alpaca-7b | 7 B | 32 |
| vicuna-7b | 7 B | 32 |
| llama-13b | 13 B | 40 |
| vicuna-13b | 13 B | 40 |
| stable-vicuna-13b | 13 B | 40 |
| llama-33b | 33 B | 60 |
| vicuna-33b | 33 B | 60 |
| gpt2-small | 124 M | 12 |
| gpt2-small-alpaca | 124 M | 12 |
| gpt2-medium | 355 M | 24 |
| gpt2-medium-alpaca | 355 M | 24 |
| gpt2-large | 774 M | 36 |
| gpt2-large-alpaca | 774 M | 36 |
| gpt2-xl | 1.5 B | 48 |
| gpt2-xl-alpaca | 1.5 B | 48 |

Table 3: **Parameter count and number of layers for all vanilla and instruction-tuned LLMs.** For the parameter count, "M" refers to million and "B" refers to billion. The number of layers for T5 models is a sum of the number of encoder and decoder layers.

# B    Language Models: Links to models weights

| Model | Link to model weights |
|---|---|
| t5-small | www.huggingface.co/google/t5-v1_1-small |
| flan-t5-small | www.huggingface.co/google/flan-t5-small |
| t5-base | www.huggingface.co/google/t5-v1_1-base |
| flan-t5-base | www.huggingface.co/google/flan-t5-base |
| flan-alpaca-base | www.huggingface.co/declare-lab/flan-alpaca-base |
| t5-large | www.huggingface.co/google/t5-v1_1-large |
| flan-t5-large | www.huggingface.co/google/flan-t5-large |
| flan-alpaca-large | www.huggingface.co/declare-lab/flan-alpaca-large |
| t5-xl | www.huggingface.co/google/t5-v1_1-xl |
| flan-t5-xl | www.huggingface.co/google/flan-t5-xl |
| flan-alpaca-xl | www.huggingface.co/declare-lab/flan-alpaca-xl |
| flan-gpt4all-xl | www.huggingface.co/declare-lab/flan-gpt4all-xl |
| flan-sharegpt-xl | www.huggingface.co/declare-lab/flan-sharegpt-xl |
| flan-alpaca-gpt4-xl | www.huggingface.co/declare-lab/flan-alpaca-gpt4-xl |
| t5-xxl | www.huggingface.co/google/t5-v1_1-xxl |
| flan-t5-xxl | www.huggingface.co/google/flan-t5-xxl |
| flan-alpaca-xxl | www.huggingface.co/declare-lab/flan-alpaca-xxl |
| llama-7b | www.github.com/facebookresearch/llama |
| alpaca-7b | www.github.com/tatsu-lab/stanford_alpaca |
| vicuna-7b | www.huggingface.co/lmsys/vicuna-7b-v1.3 |
| llama-13b | www.github.com/facebookresearch/llama |
| vicuna-13b | www.huggingface.co/lmsys/vicuna-13b-v1.3 |
| stable-vicuna-13b | www.huggingface.co/CarperAI/stable-vicuna-13b-delta |
| llama-33b | www.github.com/facebookresearch/llama |
| vicuna-33b | www.huggingface.co/lmsys/vicuna-33b-v1.3 |
| gpt2-small | https://huggingface.co/openai-community/gpt2 |
| gpt2-small-alpaca | https://huggingface.co/vicgalle/gpt2-alpaca |
| gpt2-medium | https://huggingface.co/openai-community/gpt2-medium |
| gpt2-medium-alpaca | https://huggingface.co/linkanjarad/GPT2-Medium-Alpaca-355m |
| gpt2-large | https://huggingface.co/openai-community/gpt2-large |
| gpt2-large-alpaca | https://huggingface.co/reasonwang/gpt2-large-alpaca |
| gpt2-xl | https://huggingface.co/openai-community/gpt2-xl |
| gpt2-xl-alpaca | https://huggingface.co/Rachneet/gpt2-xl-alpaca |

Table 4: **Link to model weights for all vanilla and instruction-tuned LLMs.** We provide these links for reproducibility purposes.

## C   Method for computing Brain alignment

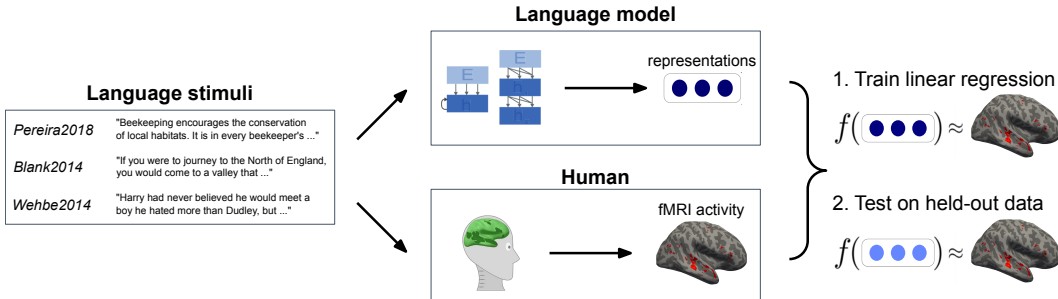

Figure 4: **Method for computing Brain alignment**, the similarity of an LLM's internal representations to human brain activity.

We follow a general approach for aligning LLM representations to fMRI brain activity used in prior works (Jain & Huth, 2018; Toneva & Wehbe, 2019; Schrimpf et al., 2021). The approaches used for the three neural datasets are similar. Hence, we explain the main details only for WEHBE2014, followed by key differences for the other two datasets.

### C.1   WEHBE2014

WEHBE2014 (Wehbe et al., 2014): The data includes fMRI recordings of 8 human participants reading chapter 9 of the book *Harry Potter and the Sorceror's Stone* (Rowling et al., 1998). Participants read the chapter at a fixed interval of one word every 0.5 seconds. The chapter contains 5,176 words.

**LLM representations.** We extract LLM representations of the text at each layer $\in \mathbb{R}^{w \times d}$, where $w$ is the number of words in the text and $d$ is the embedding size of the LLM.

**fMRI activity.** Data analyses were performed on fMRI BOLD signals extracted from the language network for all three neural datasets. fMRI activity was recorded at regular intervals of 2 seconds, with a total of 1,351 recorded time-points (TRs, times of repetition). We construct a stimulus-response matrix for fMRI activity $\in \mathbb{R}^{TR \times v_i}$, where $TR$ is the number of TRs and $v_i$ is the number of voxels in the brain of participant $i$.

**Alignment of LLM representations to fMRI activity.** We use PCA to reduce the dimensionality of the LLM representations to result in a matrix $\in \mathbb{R}^{w \times 10}$. As the number of words in the text is greater than the number of TRs, we down-sample the LLM word-level representations to the TR rate by averaging the LLM representations to the corresponding TRs, producing a matrix $\in \mathbb{R}^{TR \times 10}$. As the response recorded by fMRI peaks about 6 seconds after stimulus onset, we follow prior methods by including preceding time-points for each time-point. The final LLM representations are constructed by concatenating, for each TR, the LLM representations corresponding to the previous 4 TRs, producing a matrix $\in \mathbb{R}^{TR \times 40}$. This accounts for the lag in the hemodynamic response that fMRI records. Finally, we learn a linear function, regularized by the ridge penalty, that uses the modified LLM representations $\in \mathbb{R}^{TR \times 40}$ to predict the fMRI activity of participants $\in \mathbb{R}^{TR \times v_i}$. We train the function using 4-fold cross-validation, where each fold corresponds to a separate run of fMRI collection. We also remove 10 TRs from the LLM representations and fMRI activity from the ends of each test run to avoid train-test overlap. To compute the brain alignment, we evaluate the Pearson correlation (r) between the predicted fMRI activity $\in \mathbb{R}^{TR \times v_i}$ and recorded fMRI activity $\in \mathbb{R}^{TR \times v_i}$.

### C.2   BLANK2014

BLANK2014 (Blank et al., 2014): The data consists of fMRI recordings of 5 human participants listening to 8 naturalistic stories from the Natural Stories Corpus (Futrell et al., 2018).

fMRI activity was recorded at regular intervals of 2 seconds, with a total of 1,317 TRs. We average the BOLD signals across voxels within each language-responsive region of interest (ROI) of each participant to increase the signal-to-noise ratio, following Schrimpf et al. (2021). As there are 60 ROIs in total across the 5 participants, this produces a stimulus-response matrix $\in \mathbb{R}^{1,317 \times 60}$.

### C.3 PEREIRA2018

PEREIRA2018 (experiments 2 and 3 from Pereira et al., 2018): In experiment 2, 9 participants read 384 sentences taken from 96 text passages. In experiment 3, 6 participants read 243 sentences from 72 text passages. Each sentence was displayed for 4 seconds on a screen.

We average the fMRI responses for each sentence, resulting in one data point per sentence per language-responsive voxel of each participant. For experiment 2, there are 384 sentences and 12,195 language-responsive voxels across the participants. For experiment 3, there are 243 sentences and 8,121 language-responsive voxels across the participants. We concatenate the fMRI responses across sentences and participants for each experiment. This produces a stimulus-response matrix $\in \mathbb{R}^{384 \times 12,195}$ for experiment 2 and a stimulus-response matrix $\in \mathbb{R}^{243 \times 8121}$ for experiment 3.

## D  Method for computing Behavioral alignment

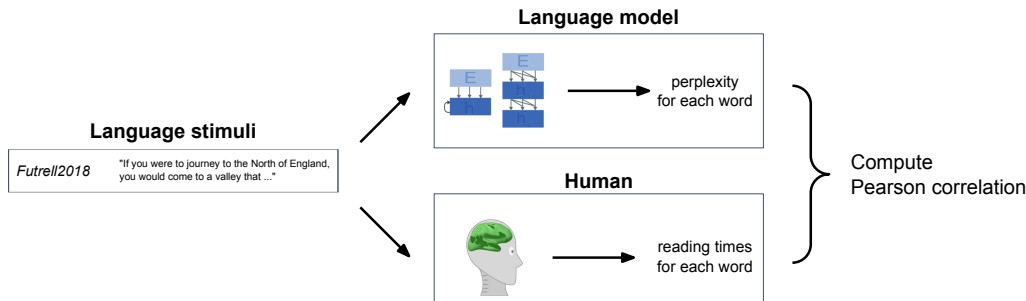

Figure 5: **Method for computing Behavioral alignment.** The same language stimuli are presented to LLMs and human participants, using the FUTRELL2018 benchmark in Brain-Score, which contains naturalistic stories. We compute the behavioral alignment as the Pearson correlation between LLM perplexity for each word and human reading times for each word.

FUTRELL2018 (Futrell et al., 2018): The data consists of reading times (RTs) of 179 participants on a total of 10,256 words across 485 sentences, taken from 10 stories from the Natural Stories Corpus (Futrell et al., 2018). The stories were presented online to Amazon Mechanical Turk users visually in a dashed moving window display. Participants press a button to reveal each consecutive word. When they press the button, the most recent word is converted back to dashes, and the next word is revealed. The time between button presses offers an estimate of comprehension difficulty (Ehrlich & Rayner, 1981; Hale, 2001; Smith & Levy, 2013). We construct a stimulus-response matrix of RTs $\in \mathbb{R}^{10,256 \times 179}$.

We provide the same stimuli to an LLM and record its perplexity for each word, producing an LLM perplexity matrix $\in \mathbb{R}^{10,256}$. We compute behavioral alignment as the Pearson correlation between per-word LLM perplexity and human RTs. Perplexity for LLMs and reading times for humans offer insights into comprehension difficulty (Ehrlich & Rayner, 1981; Hale, 2001; Smith & Levy, 2013), allowing us to examine whether LLMs and humans share similarities in terms of which words and sentences they find challenging or surprising.

# E  MMLU and BBH benchmarks

MMLU is designed to measure knowledge from many domains Hendrycks et al. (2021). It contains 57 tasks, categorized by the subject domain of world knowledge tested: STEM, Humanities, Social Sciences, and Others. The STEM category includes questions on computer science, physics, mathematics, etc. The Humanities category includes questions on philosophy, law, history, etc. The Social Sciences category includes questions on politics, sociology, economics, geography, etc. The Others category includes questions on business topics such as finance, accounting, as well as general knowledge of global facts.

BBH is designed to evaluate various problem-solving and reasoning abilities of LLMs (Suzgun et al., 2022). BBH contains 23 tasks, categorized by the type of problem-solving ability tested: (1) Algorithmic and Multi-Step Arithmetic Reasoning, (2) Natural Language Understanding, (3) Use of World Knowledge, and (4) Multilingual Knowledge and Reasoning. The world knowledge category of BBH contains tasks that test for factual and general knowledge. Tasks requiring factual knowledge include: "Sports Understanding" and "Movie Recommendation". Tasks requiring general knowledge include: "Causal Judgement", which tests knowledge about causal-reasoning suppositions, and "Ruin Names", which requires knowledge about human perception and usage of humor in the English language.

For both benchmarks, we adopt the same category classification as used in their original papers. We measure the performance of LLMs on BBH and MMLU using the `instruct-eval` repository[3] with default settings (3-shots, 5-shots respectively) and preset prompts.

# F  Code Repositories

We use the Brain-Score repository to evaluate brain alignment for the PEREIRA2018 and BLANK2014 datasets, as well as behavioral alignment for the FUTRELL2018 dataset. Link: `www.github.com/brain-score/language`.

We use an open-source repository to evaluate brain alignment for the WEHBE2014 dataset. Link: `www.github.com/awwkl/brain_language_summarization`, which builds on `www.github.com/mtoneva/brain_language_nlp`.

We use the Instruct-Eval repository to evaluate MMLU and BBH scores. Link: `www.github.com/declare-lab/instruct-eval`.

We use the Stanford Alpaca repository for instruction-tuning. Link: `www.github.com/tatsu-lab/stanford_alpaca`).

---

[3]https://github.com/declare-lab/instruct-eval

# G  Results for Brain alignment

| | PEREIRA2018 | BLANK2014 | WEHBE2014 | Average |
|---|---|---|---|---|
| t5-small | 0.166 | 0.168 | 0.071 | 0.135 |
| flan-t5-small | 0.202 | 0.178 | 0.079 | 0.153 |
| t5-base | 0.222 | 0.188 | 0.074 | 0.162 |
| flan-t5-base | 0.234 | 0.178 | 0.076 | 0.163 |
| flan-alpaca-base | 0.227 | 0.179 | 0.076 | 0.161 |
| t5-large | 0.270 | 0.082 | 0.071 | 0.141 |
| flan-t5-large | 0.311 | 0.104 | 0.080 | 0.165 |
| flan-alpaca-large | 0.322 | 0.126 | 0.082 | 0.177 |
| t5-xl | 0.285 | 0.192 | 0.072 | 0.183 |
| flan-t5-xl | 0.314 | 0.215 | 0.072 | 0.200 |
| flan-alpaca-xl | 0.312 | 0.209 | 0.075 | 0.199 |
| flan-gpt4all-xl | 0.300 | 0.206 | 0.078 | 0.195 |
| flan-sharegpt-xl | 0.323 | 0.211 | 0.070 | 0.201 |
| flan-alpaca-gpt4-xl | 0.302 | 0.205 | 0.073 | 0.193 |
| t5-xxl | 0.343 | 0.297 | 0.096 | 0.246 |
| flan-t5-xxl | 0.350 | 0.268 | 0.103 | 0.240 |
| flan-alpaca-xxl | 0.346 | 0.268 | 0.102 | 0.239 |
| llama-7b | 0.405 | 0.154 | 0.118 | 0.226 |
| alpaca-7b | 0.420 | 0.167 | 0.118 | 0.235 |
| vicuna-7b | 0.399 | 0.152 | 0.119 | 0.223 |
| llama-13b | 0.412 | 0.133 | 0.115 | 0.220 |
| vicuna-13b | 0.423 | 0.148 | 0.116 | 0.229 |
| stable-vicuna-13b | 0.415 | 0.144 | 0.115 | 0.225 |
| llama-33b | 0.426 | 0.145 | 0.109 | 0.227 |
| vicuna-33b | 0.436 | 0.156 | 0.105 | 0.232 |
| gpt2-small | 0.305 | 0.123 | 0.088 | 0.172 |
| gpt2-small-alpaca | 0.298 | 0.121 | 0.081 | 0.167 |
| gpt2-medium | 0.329 | 0.081 | 0.089 | 0.166 |
| gpt2-medium-alpaca | 0.325 | 0.080 | 0.090 | 0.165 |
| gpt2-large | 0.342 | 0.077 | 0.101 | 0.173 |
| gpt2-large-alpaca | 0.336 | 0.074 | 0.101 | 0.170 |
| gpt2-xl | 0.358 | 0.140 | 0.102 | 0.200 |
| gpt2-xl-alpaca | 0.343 | 0.139 | 0.093 | 0.192 |

Table 5: **Brain alignment results for all vanilla and instruction-tuned LLMs.** We provide these results for reproducibility purposes.

| | PEREIRA2018 | BLANK2014 | WEHBE2014 | Average |
|---|---|---|---|---|
| Noise ceiling | 0.359 | 0.210 | 0.104 | 0.224 |

Table 6: **Noise ceiling estimates for all 3 neural datasets.** fMRI measurements inherently include noise, i.e., fluctuations not due to neurons firing, from sources such as environmental factors, the scanner, and movements of human participants. Consequently, a "noise ceiling" is often computed by recording multiple samples from the same subjects for the same stimuli, providing an upper-limit estimate of the possible correlation in fMRI activity. In our study, we followed prior work for calculating the noise ceiling. For PEREIRA2018 and BLANK2014, noise ceiling estimates are computed using the Brain-Score repository, with details provided in Schrimpf et al. (2021). For WEHBE2014, noise ceiling estimates are computed using a similar procedure.

# H   Results for Next-word prediction, MMLU, BBH

|  | WikiText-2 NWP Loss | MMLU Score | BBH Score |
|---|---|---|---|
| flan-t5-small | 0.851 | 0.294 | 0.287 |
| flan-t5-base | 1.235 | 0.341 | 0.308 |
| flan-alpaca-base | 1.074 | 0.304 | 0.266 |
| flan-t5-large | 0.625 | 0.419 | 0.370 |
| flan-alpaca-large | 0.648 | 0.397 | 0.276 |
| flan-t5-xl | 0.650 | 0.493 | 0.402 |
| flan-alpaca-xl | 0.604 | 0.466 | 0.270 |
| flan-gpt4all-xl | 0.625 | 0.337 | 0.212 |
| flan-sharegpt-xl | 0.664 | 0.446 | 0.363 |
| flan-alpaca-gpt4-xl | 0.593 | 0.456 | 0.348 |
| flan-t5-xxl | 0.638 | 0.545 | 0.443 |
| flan-alpaca-xxl | 0.607 | 0.508 | 0.229 |
| alpaca-7b | 4.201 | 0.404 | 0.328 |
| vicuna-7b | 4.387 | 0.472 | 0.331 |
| vicuna-13b | 4.130 | 0.521 | 0.387 |
| stable-vicuna-13b | 4.623 | 0.495 | 0.380 |
| vicuna-33b | 3.940 | 0.590 | 0.426 |
| gpt2-small-alpaca | 4.193 | 0.270 | 0.273 |
| gpt2-medium-alpaca | 3.333 | 0.260 | 0.276 |
| gpt2-large-alpaca | 3.614 | 0.260 | 0.278 |
| gpt2-xl-alpaca | 3.283 | 0.270 | 0.284 |

Table 7: **WikiText-2 NWP loss, MMLU Overall Score, and BBH Overall Score for all instruction-tuned LLMs.** Results for vanilla LLMs are not shown as they are not adapted for the question formats in the MMLU and BBH benchmarks. Results in gray are close to random performance. We provide these results for reproducibility purposes.

**Notes on comparing next-word prediction (NWP) loss across model families.**   The T5 and LLaMA models belong to separate model families. We wish to caution that comparing next-word prediction loss across different model families may not be meaningful. This is due to several reasons related to architectural differences, training methodologies, and objectives. (1) Architecture: T5 models have an encoder-decoder architecture while LLaMA models have a decoder-only architecture. (2) Training Objectives: The T5 models were trained on supervised and unsupervised tasks, while the LLaMA models were trained only on unsupervised text (Section 3). (3) Loss computation: The loss functions for both model families are computed differently, making it inappropriate to directly compare their loss values. (4) Evaluation Metrics: Next-word prediction loss is just one metric, and it may not capture the overall language understanding capabilities of a model. Hence, we additionally evaluate these LLMs' alignment to human brain activity, as well as their performance on problem-solving abilities (BBH) and tasks requiring world knowledge (MMLU). In summary, while NWP loss is a valuable metric for evaluating language models within the same family or architecture, comparing across different model families may not be meaningful.

# I Results for Correlations of Brain Alignment with LLM properties

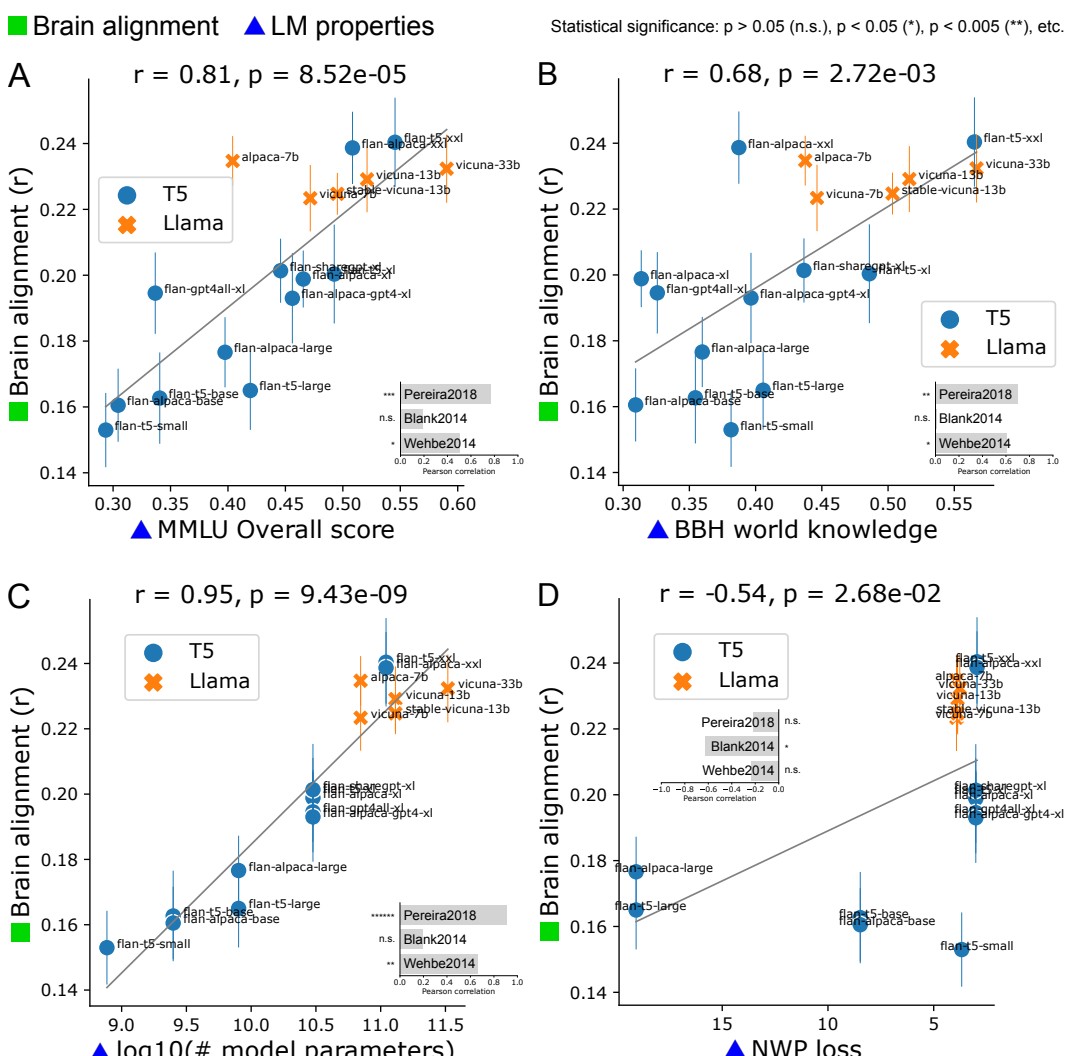

Figure 6: **Correlation between brain alignment and various LLM properties:** (A) MMLU benchmark overall score, (B) BBH benchmark score for world knowledge tasks, (C) number of parameters of the model, and (D) Next-word prediction (NWP) loss.

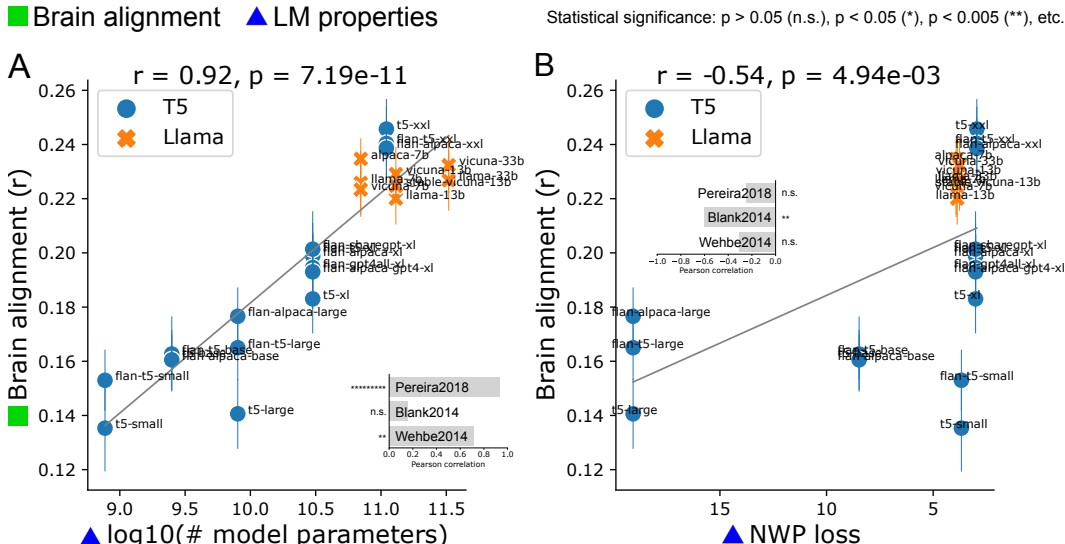

Figure 7: **Correlation between brain alignment and various LLM properties for all LLMs (including vanilla models):** (A) number of parameters of the model, and (B) Next-word prediction (NWP) loss.

## J   Additional Brain Alignment results (Linear, RSA, CKA) for Gemma and LLaMA-2 models

| Pretrained models | | | | Instruction-tuned models | | | |
|---|---|---|---|---|---|---|---|
| **Model** | **Linear** | **CKA** | **RSA** | **Model** | **Linear** | **CKA** | **RSA** |
| Gemma-2B | 0.76 | 1.94 | 0.63 | Gemma-2B | 0.72 | 2.44 | 0.70 |
| Gemma-7B | 0.90 | 2.25 | 0.62 | Gemma-7B | 0.94 | 2.55 | 0.72 |
| LLaMA-2-7B | 0.97 | 1.81 | 0.65 | LLaMA-2-7B | 1.02 | 2.00 | 0.63 |
| LLaMA-2-13B | 1.08 | 2.51 | 0.68 | LLaMA-2-13B | 1.08 | 2.77 | 0.65 |

Table 8: **Additional Brain Alignment results (Linear, RSA, CKA) for Gemma and LLaMA-2 models.** "Linear" refers to the linear predictivity metric used in the main paper, normalized by the noise ceiling computed in Appendix G.

In Section 4, we study the representational similarity between LLMs and human brain activity (brain alignment) using the linear predictivity similarity metric ("Linear"). This metric involves training a linear regression model to predict fMRI brain activity based on LLM representations. Here, we additionally validate our results using the RSA and CKA similarity metrics (Table 8), confirming that the brain alignment results are consistent across different similarity metrics.

In Section 4.1, we observed that instruction-tuning generally improves brain alignment for models from the T5 and LLaMA-1 families. We further validate this trend with recently released models from the Gemma and LLaMA-2 families (Table 8). These results hold across various similarity metrics, including Linear predictivity, RSA, and CKA.

# K    Results for Instruction-tuning LLaMA-7B on Alpaca dataset

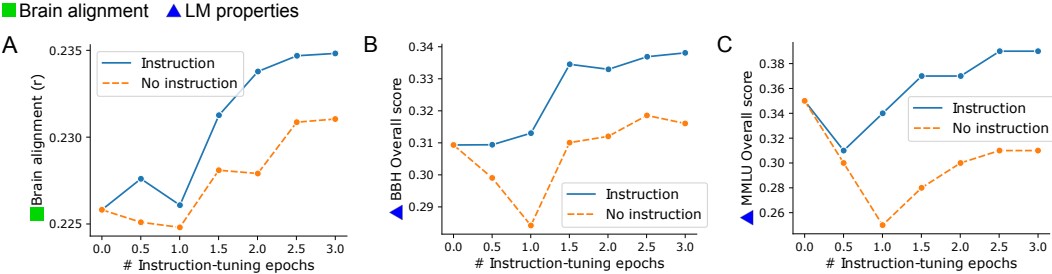

Figure 8: **Improvements in brain alignment from instruction-tuning are due to both additional training data, as well as training to understand and follow instructions.**

**Instruction model.**    We instruction-tune LLaMA-7B on the Stanford Alpaca dataset (Taori et al., 2023) using the default training process, following the code in `www.github.com/tatsu-lab/stanford_alpaca`. In particular, the model is instruction-tuned using 52K instruction-following examples generated through methods inspired by Self-Instruct (Wang et al., 2022a). This model is labeled "Instruction" in Figure 8.

**No instruction model (Ablation).**    We also train an ablation model with the same process and training data as the default instruction-tuning, but remove the instruction portion from each training sample. This ablation model is labeled "No instruction" in Figure 8. This ablation experiment disentangles: (1) training data (present in both "Instruction" and "No instruction"), from (2) training LLMs to understand and follow instructions (present only in "Instruction").

We use all provided training samples from the Alpaca dataset, thus ensuring that the models are trained on the same amount of data. We observe that brain alignment of the "No Instruction" ablation model increases during fine-tuning but stays lower than its "Instruction" counterpart. This shows that brain alignment improvements are due to both (1) training data (present in both models) and (2) the process of training LLMs to understand and follow instructions (present only in the "Instruction" model).

## L   Results for Behavioral alignment

| | FUTRELL2018 |
|---|---|
| t5-small | 0.229 |
| flan-t5-small | 0.054 |
| t5-base | 0.333 |
| flan-t5-base | 0.152 |
| flan-alpaca-base | 0.290 |
| t5-large | 0.303 |
| flan-t5-large | 0.145 |
| flan-alpaca-large | 0.291 |
| t5-xl | 0.225 |
| flan-t5-xl | 0.113 |
| flan-alpaca-xl | 0.181 |
| flan-gpt4all-xl | 0.251 |
| flan-sharegpt-xl | 0.285 |
| flan-alpaca-gpt4-xl | 0.250 |
| t5-xxl | 0.260 |
| flan-t5-xxl | 0.274 |
| flan-alpaca-xxl | 0.267 |
| llama-7b | 0.204 |
| alpaca-7b | 0.206 |
| vicuna-7b | 0.205 |
| llama-13b | 0.184 |
| vicuna-13b | 0.184 |
| stable-vicuna-13b | 0.196 |
| llama-33b | 0.158 |
| vicuna-33b | 0.164 |
| gpt2-small | 0.367 |
| gpt2-small-alpaca | 0.335 |
| gpt2-medium | 0.350 |
| gpt2-medium-alpaca | 0.345 |
| gpt2-large | 0.331 |
| gpt2-large-alpaca | 0.338 |
| gpt2-xl | 0.318 |
| gpt2-xl-alpaca | 0.336 |

Table 9: **Behavioral alignment results for all vanilla and instruction-tuned LLMs.** We provide these results for reproducibility purposes.

| | FUTRELL2018 |
|---|---|
| Noise ceiling | 0.858 |

Table 10: **Noise ceiling estimates for the FUTRELL2018 reading-times dataset.** Noise ceiling estimates are computed using the Brain-Score repository, with details provided in Schrimpf et al. (2021).

## M Results for Correlations of Behavioral Alignment with LLM properties

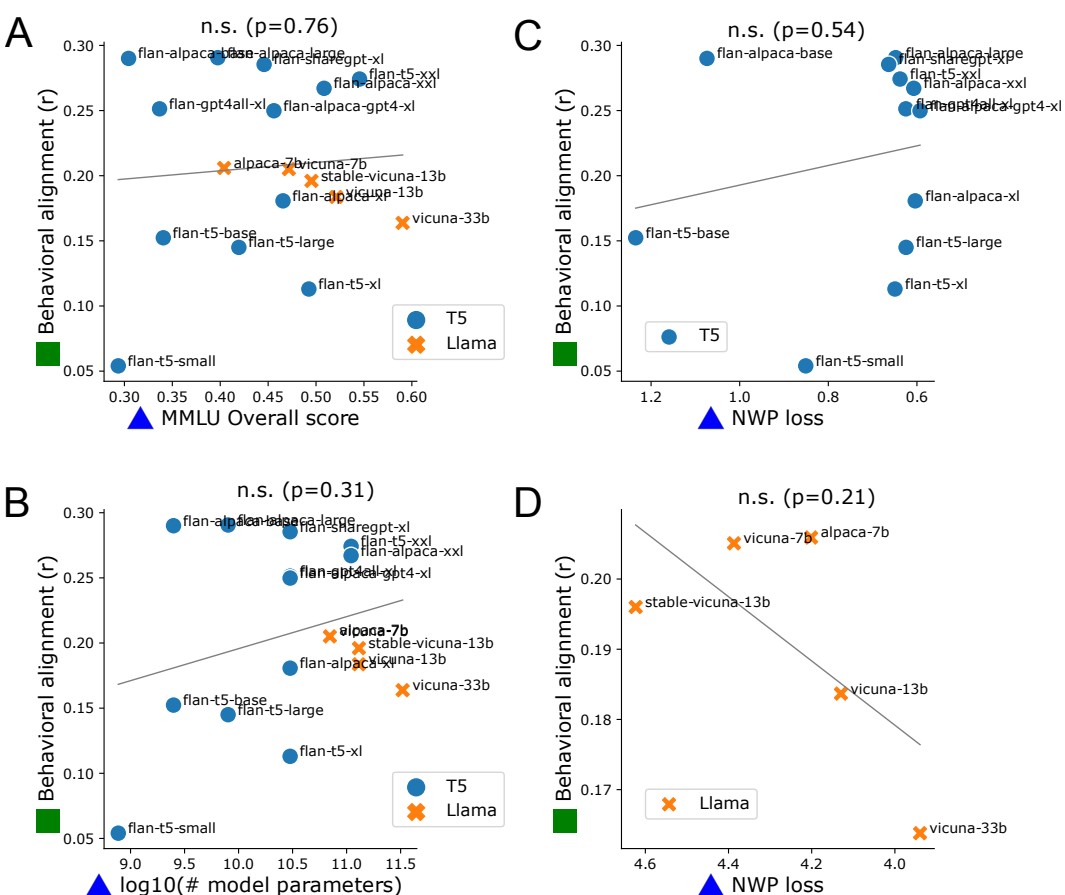

Figure 9: **Correlation between behavioral alignment and various LLM properties:** (A) MMLU benchmark overall score, (B) number of parameters of the model, (C) Next-word prediction (NWP) loss for T5 models, and (D) NWP loss for LLaMA models.

