# OpenReview forum: "Instruction-tuning Aligns LLMs to the Human Brain"
_colmweb.org/COLM/2024/Conference — COLM_

### Official Review · Reviewer_xjE7 · 2024-04-22

**Rating:** 6
**Confidence:** 3
**Ethics Flag:** 1

**Summary:**

This paper aims to study the difference between base and instruction-tuned models in terms of their alignment to both human brain and behaviours. 1) The brain alignment is measured by how well a linear regressor can predict fMRI activity from an LLM's hidden representation. 2) The behavioural alignment is measured by how well per-word perplexity correlates with per-word reading time. The work further studied the how well brain and behavioural alignment correlates with several test set and model properties.

**Reasons To Accept:**

- This paper is interesting to read and contributes knowledge to the field of HCI and LLM-humanalignment. I find this task/setting to be relatively novel.
- Technically, the work presented a correlation analysis at two levels:
  - how base/instruction-tuning affects brain and behaviour alignment.
  - how test set (knowledge being tested) and model (size, capability) properties then correlate with the brain and behaviour alignment coefficients.

**Reasons To Reject:**

Major:
- From the results in the brain alignment part, I agree that instruction tuning aligns "better" than base/pre-trained models, but the Pearson correlation coefficients are still in the "no correlation" to "weak positive" range. If I did not misunderstood: the average improvement (from base to instruction tuning) seems significant, but it is a % score derived from the difference between already weak correlation coefficients. I think that a quantitative definition of "alignment" should be provided or explained, e.g. how good the correlation needs to be in order to qualify for "alignment"; otherwise it is unsound to claim the title of this paper.
- The introduction states that "to investigate why instruction-tuning increases alignment to human brain activity [...]" which formulates the problem as a causal analysis, but the presented methodology is a post-hoc correlation study. I fear that the conclusion might lack generalisability.

Minor:
- Slight overclaim in introduction "however, there has been no similar study on how instruction-tuning". [1] has done an analysis from the perspective of LLM and human attention.
- The methodology used for behavioural alignment is "the similarity between LLM per-word perplexity and human per-word reading times", which is an accepted method but the perplexity and reading time should not be used to define behavioural alignment. ~~In fact the features used in [1] yielded different conclusions.~~ (the reviewer made an incorrect statement in the original review.)

[1] Roles of Scaling and Instruction Tuning in Language Perception: Model vs. Human Attention (Gao et al., EMNLP Findings 2023)

---

> ### Author Rebuttal · Authors · 2024-05-31
>
> We appreciate that the reviewer found our paper interesting to read, contributes knowledge to HCI and LLM-human alignment, and the task/setting is relatively novel.
>
> 1. Brain alignment of LLMs is weakly positive
>
> The reviewer is concerned about the weakly positive correlations in our brain alignment results. However, some of our LLMs actually obtain brain alignment close to the “noise ceiling” (Appendix G), suggesting they align nearly all useful signals from fMRI data. For background, fMRI measurements include noise (i.e., fluctuations not due to neurons firing) from factors such as the environment, scanner, and movements of the subject. As a result, a “noise ceiling” is an upper-limit estimate of possible correlation with fMRI activity (usually computed using multiple samples from the same subjects for the same stimuli). There is much work to be done to improve noise ceiling estimates, but we followed prior work in its calculation.
>
> 2. The study is not a causal analysis
>
> Our experiments on instruction-tuning’s effect on brain alignment are, in fact, a causal analysis. We directly compare each vanilla LLM (e.g., LLaMA-7B) before and after instruction-tuning (e.g., Alpaca-7B) (Figure 1B), controlling for factors not linked to instruction-tuning.
>
> 3. Related work on instruction-tuning [1]
>
> Thanks for the relevant pointer! Our study is complementary to theirs. We focus more on brain alignment, using 3 neural and 1 behavioral dataset (while they only touch on brain alignment in the appendix), use human reading times for behavioral alignment (while they use human attention), and study LLaMA, T5, and GPT2 families (while they only explore LLaMA).
>
> 4. Behavioral alignment method should not be used, and yields different conclusions from [1]
>
> Our work actually yields a similar conclusion to [1] and [3] for behavioral alignment: improving LLM performance does not enhance alignment with human reading behavior (Figure 3A). [1] study LLM attention and human reading attention while we study LLM surprisal and human reading times, same as [2-3].
> Moreover, human reading time is commonly used in the literature to measure behavior alignment [2-3].
>
> [1] Roles of Scaling and Instruction Tuning in Language Perception: Model vs. Human Attention
>
> [2] The neural architecture of language: Integrative modeling converges on predictive processing
>
> [3] Large GPT-like Models are Bad Babies: A Closer Look at the Relationship between Linguistic Competence and Psycholinguistic Measures

---

> ### Author Response · Authors · 2024-06-04
>
> Thank you once again for taking the time to review our work!
> We look forward to receiving your feedback on our responses to your initial review. We welcome any additional questions or discussion to ensure your outstanding concerns regarding our submission are thoroughly addressed before the discussion period ends on June 6th.

---

> ### Comment · Reviewer_xjE7 · 2024-06-07
> **Reviewer xjE7's response after rebuttal**
>
> Thank you to the authors for your rebuttal!
>
> Re 1, 3, and 4: Thank you for your explanation and clarification. I think my questions are addressed.
>
> Re 2: I meant to say that that the "*why*" part in the statement is not well-explored.

---

### Official Review · Reviewer_hcZZ · 2024-05-09

**Rating:** 4
**Confidence:** 4
**Ethics Flag:** 1

**Summary:**

This paper is an investigation into whether instruction-tuning of LLMs affects the alignment between intermediate LLM representations and brain activity as measured by fMRI.  Alignment with brain representations is measured using the Brain-Score linear predictivity measure from prior work, and this measure is computed across a variety of language models both with and without instruction-tuning.  The experiments find that instruction tuning increases brain alignment by 6.2% on average across different base and instruction-tuned model pairs.  In addition, the brain alignment of the different LLMs is also compared with performance on benchmarks measuring world knowledge and reasoning, and a strong correlation is found with world knowledge benchmark performance but not with reasoning.  Finally, there is also an experiment in the paper comparing "behavioral alignment", which looks at the correlation between per-word perplexity of LLMs and the human per-word reading times, but does not find any significant effect of instruction tuning on this metric.

**Reasons To Accept:**

The finding that brain alignment correlates better with benchmarks measuring world knowledge than it does with benchmarks measuring reasoning is interesting.  Though as the authors note, this finding could be different if the subjects whose brain activity was being measured were performing tasks requiring more reasoning rather than reading, and the correlation with world knowledge benchmarks was still less than with model size, which has a very strong correlation that has also been seen in prior work.

**Reasons To Reject:**

This paper does not introduce new methods of analysis, it runs existing implementations of LLM-brain alignment measurement across a different set of input LLMs.

As the paper notes, much prior work has found that LLM-brain alignment is well correlated with improved LLM benchmark performance, and since instruction tuning is known to substantially improve benchmark performance it seems unsurprising that it would also follow this trend and improve alignment.

---

> ### Author Rebuttal · Authors · 2024-05-31
>
> We appreciate that the reviewer found our finding—that brain alignment correlates better with world knowledge than reasoning—interesting.
>
> **Response to concerns:**
>
> 1. No new methodology
>
> We respectfully disagree this is valid grounds for rejection. Developing new methods is one of many forms of scientific contribution; our analysis of instruction-tuning and LLM world knowledge contributes new insights to the field. From the COLM review guide: “Our goal as researchers is not only to develop methods and build artifacts but also to understand both the methods we use and natural language.” However, we understand these concerns and will update the paper to better highlight our contributions.
>
> 2. Unsurprising that instruction-tuning improves brain alignment
>
> The reviewer implies that because some methods that improve performance on certain benchmarks also increase brain alignment (though we note the absence of specific references), it is not surprising that instruction tuning would do so as well. We’re not as confident in this generalization, and we note that:
>
> (a) many of the prior works we know of that agree with the reviewer’s generalization are testing next-word prediction, not reasoning and world knowledge (which our paper is the first to study)
>
> (b) prior work has shown that finetuning may result in no change or decreased brain alignment depending on the finetuning task and brain region [1], contradicting the reviewer’s point, and
>
> (c) our work investigates novel questions. We show that instruction-tuning LLMs improves brain alignment and brain alignment correlates with better representations of world knowledge. Our results provide new insights into world knowledge representations by suggesting that mechanisms encoding world knowledge in LLMs also improve brain alignment. This opens new avenues for future research to explore how enhancing world knowledge can align LLMs to human brains. We will update the paper to make these distinctions clearer.
>
> [1] Neural Language Taskonomy: Which NLP Tasks are the most Predictive of fMRI Brain Activity? (Oota et al., 2022)

---

> ### Author Response · Authors · 2024-06-04
>
> Thank you once again for taking the time to review our work!
> We look forward to receiving your feedback on our responses to your initial review. We welcome any additional questions or discussion to ensure your outstanding concerns regarding our submission are thoroughly addressed before the discussion period ends on June 6th.

---

> ### Comment · Reviewer_hcZZ · 2024-06-05
>
> Thank you for your response.  I have made a few minor edits to the wording of my review for clarity but have decided to keep my score the same.

---

### Official Review · Reviewer_uLs2 · 2024-05-10

**Rating:** 5
**Confidence:** 4
**Ethics Flag:** 1

**Summary:**

This paper explores the correlation between instruction-tuning and brain alignment as well as behavioral alignment, demonstrating that instruction-tuning can enhance scores for brain alignment without increasing behavioral alignment scores. Additionally, the paper analyzes the correlation between brain alignment, behavioral alignment, model size, world knowledge, and model capabilities.

**Questions To Authors:**

refer to the comments

**Reasons To Accept:**

1.	The study evaluates 25 LLMs from the T5 and LLaMA families, as well as 8 models from the GPT2 family, including their original versions and various instruction-tuned versions, offering a comprehensive analysis of instruction tuning across these three model families.
2.	The paper examines the correlation between brain alignment, behavioral alignment, and the characteristics of LLMs.

**Reasons To Reject:**

1.	Figures 2 and 4 (B, C, D) are noted to be somewhat unclear, suggesting a need for refinement in their presentation for better clarity.
2.	The paper would benefit from an expanded analysis that includes models such as LLaMA2 and Mistral to more effectively demonstrate the impact of instruction tuning. Additionally, the inclusion of an analysis on the correlation between large language model alignment methods, such as RLHF, and brain alignment would contribute to the comprehensiveness of the study.
3.	It is recommended that the use of reading time in behavioral alignment be reconsidered to include the potential impact of word length. The analysis could potentially be more thorough if it differentiates between long and short words based on a defined word length threshold, and also takes into account the reading time adjusted for word length, such as by using a reading time to word length ratio.
4.	Representational Similarity Analysis (RSA) and Brain-Score can both indicate brain alignment, and the authors might explore whether RSA could lead to the same conclusions.

---

> ### Author Rebuttal · Authors · 2024-05-31
>
> We appreciate that the reviewer found our analysis of instruction-tuning (IT) comprehensive, using many LLMs from multiple model families.
>
> 1. Figures 2 and 4 are somewhat unclear
>
> We will remove model names and adapt point sizes to model sizes for better readability. We provide larger figure versions with model names in the Appendix.
>
> 2. Test LLaMA2, Mistral. Study RLHF’s effect on brain alignment
>
> Instead of testing many models with highly similar architectures, we prioritized models from different families, particularly those with varying model sizes (LLaMA, T5, GPT-2). Below are results normalized relative to the estimated noise ceiling for LLaMA2 and Gemma, showing that IT models (right score) have higher alignment than vanilla (left score).
>
> Regarding the impact of RLHF, which aligns LLMs to human preferences, on LLMs alignment to human brain activity and behavior: we studied one RLHF model (StableVicuna-13B), but our findings were insufficient to draw rigorous conclusions. Our findings are more comprehensive for IT, the focus of our paper, though we agree rigorous investigation of RLHF’s effect on brain alignment is an interesting future direction.
>
> |Model|Linear|CKA|RSA|
> |-|-|-|-|
> |Gemma-2B|.76/.72|1.94/2.44|.63/.70|
> |Gemma-7B|.90/.94|2.25/2.55|.62/.72|
> |LLaMA2-7B|.97/1.02|1.81/2.0|.65/.63|
> |LLaMA2-13B|1.08/1.08|2.51/2.77|.68/.65|
>
> 3. Effect of word length on behavioral alignment
>
> Behavioral alignment evaluates the correlation between per-word reading times and LLM per-word surprisal for the same stimuli. We follow [1] and implicitly account for word length by computing the surprisal of a word with multiple tokens as the sum of surprisals of each token. Importantly, we use consistent tokenizers when comparing vanilla LLMs and IT versions, thus controlling the effect of word length when evaluating the impact of IT on behavioral alignment.
>
> 4. Run other brain alignment methods (e.g., RSA) aside from Brain-Score linear predictivity
>
> We focused on linear predictivity following [1], but will add more comprehensive results for RSA [2] and CKA [3] to our revision. Early experiments (see table above) indicate similar patterns for LLaMA2 and Gemma models, except for RSA for LLaMA2.
>
> [1] The neural architecture of language: Integrative modeling converges on predictive processing
>
> [2] Representational similarity analysis - connecting the branches of systems neuroscience
>
> [3] Similarity of Neural Network Representations Revisited

---

> ### Author Response · Authors · 2024-06-04
>
> Thank you once again for taking the time to review our work!
> We look forward to receiving your feedback on our responses to your initial review. We welcome any additional questions or discussion to ensure your outstanding concerns regarding our submission are thoroughly addressed before the discussion period ends on June 6th.

---

> > ### Comment · Reviewer_uLs2 · 2024-06-07
> >
> > Thank you for your response. While some of the concerns have been addressed, there are still significant issues, such as the evaluation, that have not been resolved. As a result, I will maintain my current rating.

---

### Official Review · Reviewer_edNc · 2024-05-10

**Rating:** 8
**Confidence:** 4
**Ethics Flag:** 1

**Summary:**

This work investigates whether instruction-tuning can make LLMs process language similar to humans. In particular, this work studies:
Brain Alignment: How closely LLMs’ internal representations match neural activities in the human language system.
Behavioral Alignment: How similar LLM behaviors are to human behaviors during language tasks.
Comprehensive and carefully-designed experiments are conducted to draw the conclsion:

Instruction-tuning generally aligns LLM representations to human brain activity and instruction-tuning LLMs generally does not enhance behavioral alignment with human reading times.

**Reasons To Accept:**

1. Very interesting research problem. since the instruction tuning has been a de-facto choice for training LLMs, this study can provide some insights into whether instruction tuning can lead to human-level or human-like intelligence. Whatever the answer is, the research question per se is important and worth investigating.
2. The experiment setup is valid and comprehensive: four types of langauge abilities, 25 LLMs (instruction tuned or not) with different sizes.
3. Intersting and meaningful findings.

**Reasons To Reject:**

1. It would be great to provide more insights/future directions to how to build more brain-/human-intelligence-aligned LLMs.

The rest weaknesses here are more like a suggestion not weakness:
2. It would interesting to see the connections between the alignment and the training data, which will provide more insights to determine what's lacking in the current instruction tuning to build more brain-aligned LLMs.

3. I notice the behaviors of LLMs on reasoning tasks is not well aligned with human. I wonder whether it's a training data issue or model arch issue. There are reasoning-specific instruction tuning models such as Mammoth for math reasoning. It would be great if the models are specificly trained for a specific task can increase the alignment.

---

> ### Author Rebuttal · Authors · 2024-05-31
>
> We appreciate that the reviewer found our experiments comprehensive and carefully designed, our research problem very interesting, and our findings meaningful.
>
> 1. Discuss more insights to build brain-/human-intelligence-aligned LLMs
>
> We thank the reviewer for the suggestion. In our revision, we will add a subsection discussing papers with insights on LLM and human cognition [1-2] and how to build brain-aligned LLMs [3]: next-word prediction [4], data/model size [5], fine-tuning [6], semantic understanding [7], among others.
>
> 2. Study connections between brain alignment and training data
>
> We trained LLaMA-7B on the Alpaca instruction dataset and an ablated version without the instruction portion for each sample (Figure 1D, Appendix K). The ablated model increases less in brain alignment than the model tuned with instructions, suggesting that brain alignment benefits from both the training examples, but also the instruction format; this opens the way for further exploration.
>
> 3. Test reasoning-specific models (e.g., Math)
>
> The reviewer suggests measuring brain alignment for reasoning-specific datasets to elucidate why LLM performance on reasoning tasks is not well correlated with brain alignment improvement from instruction tuning.
> We limit our scope to the human language system and use datasets where participants read/listened to English text and did not engage in specific tasks related to abilities such as mathematical reasoning. Prior work tackled an fMRI dataset of participants reading code, which could be linked with models’ ability to understand and generate code [8]. However, they measure alignment with the Multiple Demand System of the brain which is known to be involved in domain-agnostic tasks like problem-solving, logic, and spatial memory tasks.
>
> [1] Dissociating language and thought in large language models: a cognitive perspective
>
> [2] Roles of Scaling and Instruction Tuning in Language Perception: Model vs. Human Attention
>
> [3] Getting aligned on representational alignment
>
> [4] The neural architecture of language: Integrative modeling converges on predictive processing
>
> [5] Scaling laws for language encoding models in fMRI
>
> [6] Divergences between Language Models and Human Brains
>
> [7] Lexical semantic content, not syntactic structure, is the main contributor to ANN-brain similarity of fMRI responses in the language network
>
> [8] Convergent Representations of Computer Programs in Human and Artificial Neural Networks

---

> > ### Comment · Reviewer_edNc · 2024-06-03
> >
> > Thanks for the reply and explanation.
> > Totally understand that you want to narrow the scope for the current submission.
> >
> > I will keep my score unchanged.

---

### Author Response · Authors · 2024-06-07
**Response to the AC**

We thank the reviewers for their feedback and the AC in advance for their meta-review.

We also thank the reviewers for identifying our paper’s strengths:
- **edNc**: comprehensive and carefully designed experiments, very interesting research problem, and meaningful findings.
- **uLs2**: comprehensive analysis of instruction-tuning, using many LLMs from multiple model families.
- **hcZZ**: interesting finding that brain alignment correlates better with world knowledge than reasoning.
- **xjE7**: interesting to read, contributes knowledge to HCI and LLM-human alignment, and the task/setting is relatively novel.

We summarize each reviewer’s suggestions/concerns and our responses:

**Reviewer edNc**:
- Suggested we discuss more insights to build brain-/human-intelligence-aligned LLMs. In our paper, we will add a subsection discussing papers with insights on LLM and human cognition and how to build brain-aligned LLMs.  (References [1-7] in our rebuttal).
- Suggested we study connections between brain alignment and training data. We explain that we instruction-tuned LLaMA-7B and an ablated version, and how that provides preliminary insights.
- Suggested we test reasoning-specific models (e.g., Math). We explain that reasoning-specific models may not be appropriate as we limit our scope to the human language system and use datasets where participants read/listened to English text.

**Reviewer uLs2**:
- Noted that two figures in our paper could be clearer. We remove model names and adapt point sizes to model sizes for better readability. We provide larger figure versions with model names in the Appendix.
- Suggested we test additional models, e.g., LLaMA2, Mistral. We explain we prioritized models with diverse model architectures and families (LLaMA, T5, GPT-2). Regardless, in our rebuttal, we show results for LLaMA2 and Gemma models, yielding similar results to our paper’s conclusion.
- Suggested we study the effect of RLHF. We explain that we studied only one RLHF model, which is insufficient to draw rigorous conclusions. Our paper’s focus is instruction-tuning, for which we provide comprehensive results.
- Asked about the effect of word length on our behavioral alignment method. We explain that we follow prior works in implicitly accounting for word length by computing the surprisal of a word with multiple tokens as the sum of surprisals of each token. Importantly, we use consistent tokenizers when comparing each vanilla LLM to its instruction-tuned version, thus controlling for the effect of word length.
- Suggested we run other brain alignment methods (e.g., RSA) aside from Brain-Score linear predictivity. In our rebuttal, we add preliminary results for RSA and CKA, showing they are similar to linear predictivity. We focused on linear predictivity following prior works, but will add more comprehensive RSA and CKA results to our paper.

**Reviewer hcZZ**:
- Concerned that our paper has no new methodology. We cite the COLM review guide in explaining that developing new methods is only one of many forms of scientific contribution; our analysis of instruction-tuning and LLM world knowledge contributes new insights to the field.
- Implied that since LLM performance is correlated to brain alignment (shown in prior work), it is expected that instruction-tuning improves brain alignment since it improves LLM performance. We explain why we are not confident in this generalization. (a) Prior works evaluate LLM performance using next-word prediction, whereas we study reasoning and world knowledge. (b) Contrary to the reviewer’s point, prior work showed that finetuning may result in no change or decreased brain alignment depending on the finetuning task and brain region.

**Reviewer xjE7**:
- Asked about the weakly positive correlations in our brain alignment results. We explain that some of our LLMs actually obtain brain alignment close to the fMRI “noise ceiling” (further explained in our rebuttal and paper), suggesting they align nearly all useful signals from fMRI data. We follow prior works in computing the noise ceiling.
- Claimed that our study is not a causal analysis. We explain how our experiments showing that instruction-tuning improves brain alignment are, in fact, a causal analysis.
- Noted a related work (reference [1] in their review) on instruction-tuning. We explain that our study is complementary to [1], with many key differences in datasets and methods. We will discuss this related work in our paper.
- Suggested our behavioral alignment method should not be used, and yields different conclusions from [1]. We explain that our behavioral alignment method is commonly used in the literature. Also, it actually yields a similar conclusion to [1]: improving LLM performance does not enhance alignment with human reading behavior.

We will add these clarifications to improve our paper. Thank you again!

-Authors

---

### Decision · Program_Chairs · 2024-07-10

**Decision:**

Accept

**Comment:**

## Summary of the paper

The paper studies how instruction tuning, i.e. supervised fine-tuning on instruction-following data, affects
- Brain alignment: predictability of human brain activity from LLM activations (when humans and LLMs read the same text)
- Behavior alignment: correlation between LLM perplexity and human reading times

The authors show a significant positive impact from instruction tuning on the Brain alignment. The authors argue that in some cases the to the results are close to the noise level of the data, i.e. optimal performance. The authors also show that brain alignment strongly correlates with performance on certain benchmarks such as MMLU.

On the behavior alignment task, the authors show negative or no improvement from instruction tuning.

## Strengths

- This is the first paper to the best of my knowledge that studies the impact of instruction tuning on LLM-Brain alignment
- The results are interesting and not obvious, and contribute to the broader picture of Brain-LLM alignment
- The authors study a broad range of models; during the rebuttal, the authors added results for the recent Gemma and Llama2 models, and the original paper already included many models
- The poor correlation of perplexity and reading times is somewhat unexpected

## Weaknesses

- I agree with reviewer uLs2 that studying RLHF as opposed to supervised finetuning on instruction data would be very interesting. The industry standard for LLM post-training is RLHF and understanding the effect of RL would be extremely interesting, especially if it is different from supervised finetuning.
- Reviewers uLs2 and xjE7 raised concerns about the behavior alignment method: e.g. possibly the reading times should be normalized by the total length of the text in characters. The results on behavior alignment are surprising, and it would be interesting to provide further insights here.
- Reviewer hcZZ pointed out that the paper does not develop new methodology, and instead applies existing methodology in a new setting. I am inclined to agree with the authors that this is not a major issue.
- I think it would be extremely interesting to push the hypothesis on correlation of Brain Alignment and model size to the limit and try models like Llama2-70, Nemotron-4 340B, and maybe Grok-1 314B.

## Conclusion

Overall, I think this is an interesting paper, and it would be a valuable contribution to the conference, despite some limitations of the analysis.

[At least one review was discounted during the decision process due to quality]